ecology/evolution

language isolates, language geography, language diversity, isolation by environment

**Author for correspondence:**
Matthias Urban
e-mail: matthias.urban@uni-tuebingen.de

# The geography and development of language isolates

## Matthias Urban

Center for Advanced Studies 'Words, Bones, Genes, Tools', University of Tübingen, Rümelinstrasse 19-23, 72070 Tübingen, Germany

 MU, 0000-0001-7633-7433

This contribution theorizes the historical dynamics of so-called language isolates, languages which cannot be demonstrated to belong to any known language family. On the basis of a qualitative review of how isolates, language families or their branches lost territory to other languages through time, I develop a simple model for the genesis of isolates as a function of proximity to major geographical barriers, and pit it against an alternative view that sees them as one manifestation of linguistic diversity generally. Using a variety of statistical techniques, I test both accounts quantitatively against a worldwide dataset of language locations and distances to geographical barriers, and find support for the position that views language isolates as one manifestation of linguistic diversity generally. However, I caution that different processes which are not necessarily mutually exclusive may have shaped the present-day distribution of language isolates. These may form elements of a broader theory of language isolates in particular and language diversity in general.

## 1. Introduction

It has almost become a truism to say that linguistic diversity is unevenly distributed across the globe. This pertains both to genealogical diversity (the number of language families, groups of languages that can be shown to have descended from a common ancestor language, in a given area), language richness (the gross number of languages in a given area, regardless of their genealogical affiliation), and also typological diversity (the degree to which languages in a given area make use of different grammatical and lexical resources to achieve their communicative functions). Since the turn of the century, interest in the social, but also the geophysical factors that underlie the observed diversity has received a major surge. In this context, a number of highly heterogeneous factors—ranging from language ideologies to terrain rugosity and a wealth of others—have been identified as generating and/or maintaining

linguistically diverse landscapes [1–14]. However, the results of different studies on the environmental drivers of language diversity are frequently incompatible with one another [6,12], and, while clearly connected [10,11,15], on global scales it is as yet difficult to link social drivers of language diversity and possible environmental factors in a meaningful way.

One class of languages that contributes significantly to overall levels of genealogical diversity are so-called language isolates. These are languages which cannot be shown by accepted methods of historical-comparative linguistics to belong to any known language family. Accordingly, each isolate forms a self-contained language family—defined as a set of genealogically related languages—the only member of which is it itself (nevertheless, in line with common usage, I will reserve the term 'family' in the remainder of this article for a set of related languages that is larger than one). At the same time, we know that the historical dynamics of languages is characterized by a complex and never-stopping ebb and flow of diversification, spread and extinction. Under this premise, it is difficult to conceive of a situation in which a language should develop through millennia in complete phylogenetic inertia (i.e. neither losing relatives nor diversifying itself into a family). Under a Normal Diachrony Assumption [16], then, we may surmise that isolates are subject to the same processes of transmission with change as other languages. Hence, they must have a history involving diversification and fission (cladogenesis in biological terms)—the only difference to languages belonging to larger families is that in the case of isolates, we do not have knowledge of the related languages because too much time has passed to recognize them or because they have ceased to be spoken. In accordance with these considerations, one common and productive way to look at language isolates is to view them as the last surviving members of a former language family [17,18]. This stance is also supported empirically by languages which currently qualify as language isolates, but which are known to have had related members, e.g. Ket, the only surviving language of the Yeniseian family of Siberia, or Chipaya, the only surviving member of the Uru-Chipaya family of the Bolivian *altiplano*.

Given that isolates are at the same time self-contained genealogical units, they contribute significantly to linguistic diversity, in particular genealogical diversity. Also, taking for granted that isolates are generated through the same general historical processes that drive language diversification generally, one expectation would be that they occur in hotspots of linguistic diversity. Indeed this is the case for places like New Guinea and South America, which are both rich in linguistic diversity generally as well as in language isolates specifically. Thus, isolates can be seen as one particular manifestation of linguistic diversity generally.

However, there is another way to look at isolates that also takes the Normal Diachrony Assumption as its starting point, but develops it further through a different line of thought: if isolates are conceived of as the sole survivors of former language families that were largely superseded by later language spreads, they invite to not only theorize the expansion and diversification of languages and language families, but also reductive processes that are the mirror image of expansion and diversification. Sometimes such processes show a 'retreat' of a language's or family's former range towards a major geographical barrier, prominently the coastline or a major mountain area with conditions that are unsuitable for permanent habitation. Typically, the areas in which a language or family survives in such scenarios are habitats that are less attractive for exploitation by humans than other areas of the former range. Consistent with this, a number of isolates are indeed spoken in such environments.

In the following §2, I develop two models for isolate genesis based on these alternative points of view. Because not much has been written on this from a comparative point of view, the model that emphasizes language reduction and the role of geophysical barriers requires more motivation and explanation. The discussion I provide is inductive and based on case studies of range reduction processes from different parts of the world and at various levels of resolution.

These models lead to conflicting predictions regarding the present-day locations of isolates, and in §3 I test these statistically against the actually observed present-day distribution of isolates. A higher-order methodological aim here is to put notions developed in earlier research on patterns of linguistic diversity, such as the distinction between spread and accretion zones [1,2], on a more explicit quantitative footing and relate synchronic distributions more firmly to the diachronic dynamics of language spread and especially reduction. Complementarily to this move from more qualitative observation to quantitative analysis, I aim to contribute to linking statistical modelling of linguistic diversity—here, specifically language isolates—to detailed observations on known historic processes. It is often the case that large-scale quantitative analyses of global datasets of linguistic diversity are oblivious of these patterns. However, they may be crucial as background information for developing descriptively adequate synchronic and diachronic models of the factors that shape synchronic linguistic distributions.

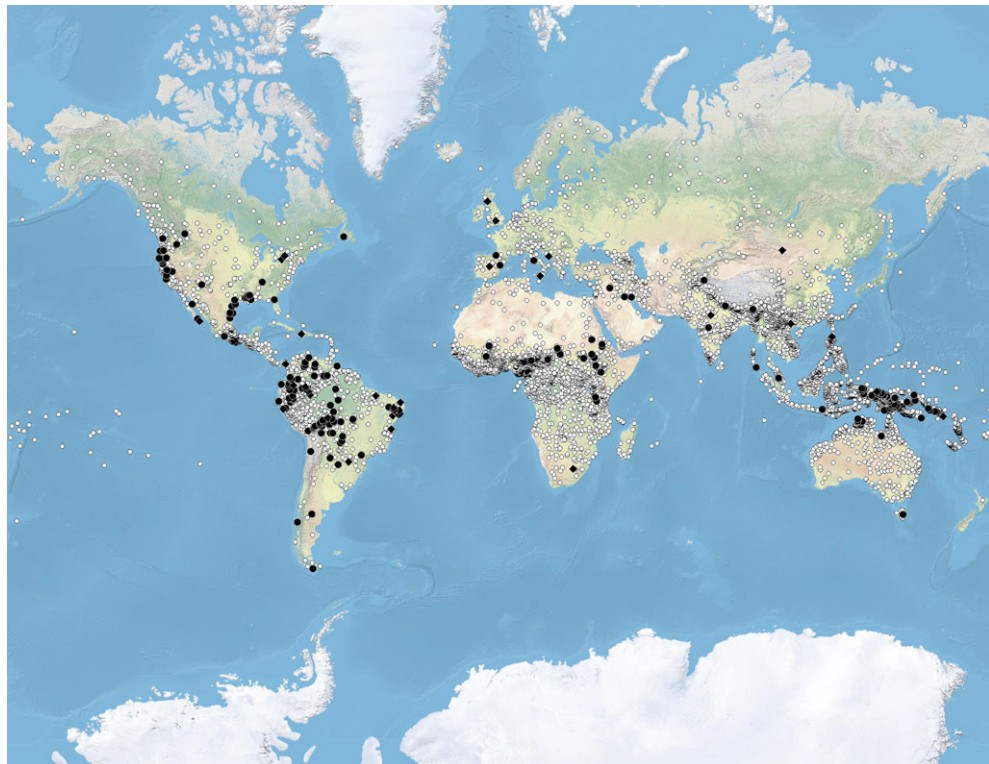

**Figure 1.** Language isolates (large black symbols) in the context of overall language diversity (small white dots). Data from Glottolog 4.2 [19]. Isolates in a narrow sense are shown by circles, additional isolates in a wider sense that include Glottolog's 'unclassified' languages are shown by black diamonds (see data and methods for discussion).

## 2. Two models for the distribution of language isolates and language diversity

A map of the world's language isolates in the context of overall linguistic diversity is in figure 1, with isolates represented by large black symbols and languages belonging to families consisting of more than one language—sometimes hundreds or even thousands—by small white symbols.

Impressionistically, in a salient number of cases, the regions where isolates cluster appear to be at the same time high diversity zones. This is true, e.g. of South America, in particular the Upper Amazon, and also of California. In Africa, likewise, isolates occur in a prominent belt south of the Sahara which is generally rich in linguistic diversity (called 'fragmentation zone' [20]). While geophysical factors may play a role in the emergence of linguistic diversity in this zone, it is not restricted to isolates. In sum, language isolates may be conceived of as one aspect of general linguistic diversity, consistent with a view on isolates that de-emphasizes their exceptionality [17]. On a very coarse macro-area level, some basic statistics support this view. Using the classification of the world's languages in Glottolog 4.2 [19], which is informed by published statements of experts working on the individual families and is generally conservative in accepting proposed genealogical relations, the statistics are as in table 1 (not counting dummy-families in Glottolog such as 'sign languages', 'bookkeeping', etc.). Using this breakdown of the world, isolates and family density are very strongly correlated at $\rho \approx 0.94$. While acknowledging that this is much too coarse a measure to be of any value beyond a first indication towards a global pattern, the correlation is significant even with the low power of the test at $p < 0.05$ by a Spearman's rank correlation.

While much recent research has been devoted to identifying the drivers of linguistic diversity on continental and global scales [6–9,12–14], these typically operate with notions such as 'group boundary formation' at a high level of abstraction that is far removed from 'on-ground' realities. Concrete qualitative explanatory accounts for the social and linguistic processes that bring about these distributions are typically not well developed, and at a meso-level, these processes indeed are often not well understood yet (though see e.g. [4,15]). One possible example for a concrete diachronic link between the distribution of isolates and general language richness and genealogical diversity comes from South America. In South America, linguistic diversity, and notably also language isolates, cluster in the western margins of

**Table 1.** Macro-areas and the number of isolates and families they contain as per Glottolog 4.2 [19].

| macro-area | no. isolates | no. language families |
|---|---|---|
| Africa | 16 | 41 |
| Australia | 9 | 26 |
| Eurasia | 12 | 31 |
| North America | 32 | 48 |
| South America | 66 | 50 |
| Papunesia | 54 | 79 |

Amazonia [21]. A 'northern diversity zone' and a 'southern diversity zone' may be distinguished [21]. The 'northern diversity zone' is at the same time one in which several westward long-distance migrations along the course of the Marañón river towards and into the Andes are in evidence [22]. One is directly attested historically in the early sixteenth century; it brought speakers of Tupinambá, who were fleeing the European invasion of their homelands and at the same time were searching for a paradisiac 'Land-Without-Evil', from coastal Brazil to the Chachapoyas area on the eastern slopes of the Andes [23]. Similar earlier movements must have brought speakers of Cocama, a far western outlier of the Tupian family, as relative newcomers to the linguistic ecology of the Amazonian tributaries and the lower courses of the Marañón specifically. Similar east-west movements in prehistory are suggested by the fact that some of the poorly documented languages of the eastern Andean slopes, Patagón and Sácata, might be related to two major language families of the Amazon, Cariban and Arawakan, respectively [24]. For a long time, something must have acted as an attractor in these regions, causing the repeated inflow of languages from the east that found an endpoint in the western margins of greater Amazonia and the eastern foothills of the Andes, thereby increasing language richness and genealogical diversity. At larger time-scales, such newcomers, isolated from the centre of gravities of their families, would have developed into languages that are not recognizably linked anymore to the language families from which they have sprung, i.e. into isolates, which is consistent with the fact that they are a prominent aspect of linguistic diversity in the region.

A second perspective on the present-day distributions of isolates emphasizes their status as possible remnants of larger families as sketched in the introduction. This perspective is less obvious when eyeballing the global present-day distributions in figure 1 and requires more qualitative information on the historical dynamics of language reduction, which I review in the following in the form of case studies to carve out commonalities.

Basque is the only language isolate of Europe that is still spoken, namely in parts of the Basque Autonomous Community of Spain and the French Pyrénées-Atlantiques department, especially the coast of the Gulf of Biscaya and its immediate hinterland (figure 2). In classical antiquity, however, the Basque-speaking region ranged from the Biscaya east to the Val d'Aran in the heart of the Pyrenees, north into the Aquitaine basin and south to the Ebro valley [25]. The present distribution reflects a constant reduction of the Basque-speaking region towards the coast. Since this process took place in historical times, it can be traced with relative ease [26,27], see figure 2.

The coastal landscape in which Basque is still spoken is characterized by limestone mountains that are incised by river valleys. While vegetation is lush owing to abundant rainfall, soils are comparably barren and difficult to cultivate, as already observed by von Humboldt [32]. This may be causally related to the reductive processes that the language underwent in historical times: 'the mountainous Basque terrain, with little agricultural land, no cities, few obvious resources, and harbours that faced uselessly (from the Roman point of view) onto the Atlantic, was simply too insignificant to be worth the trouble of colonization. And the same lack of Roman interest is very largely what guaranteed the unique survival of the Basque language' [27, p. 11] (compare [26]).

Burushaski is a language isolate that is spoken in an extremely challenging mountainous environment of northern Pakistan. Figure 3 plots the present-day distribution of Burushaski in its linguistic context.

The distribution of languages in this area is strongly predicated on the course of river valleys as 'cross-country travel is impossible and practicable routes can be found only by following the natural lines of least resistance, and that not without pain' [34, p. xxxii]. Permanent settlements are built on elevated terraces on the sides of these valleys [34] as large parts of the remaining land are uninhabitable. Furthermore, the inhabitable area is limited in the north through 'an immense mountain barrier traversable only by a few very high passes' [34, p. xxxi].

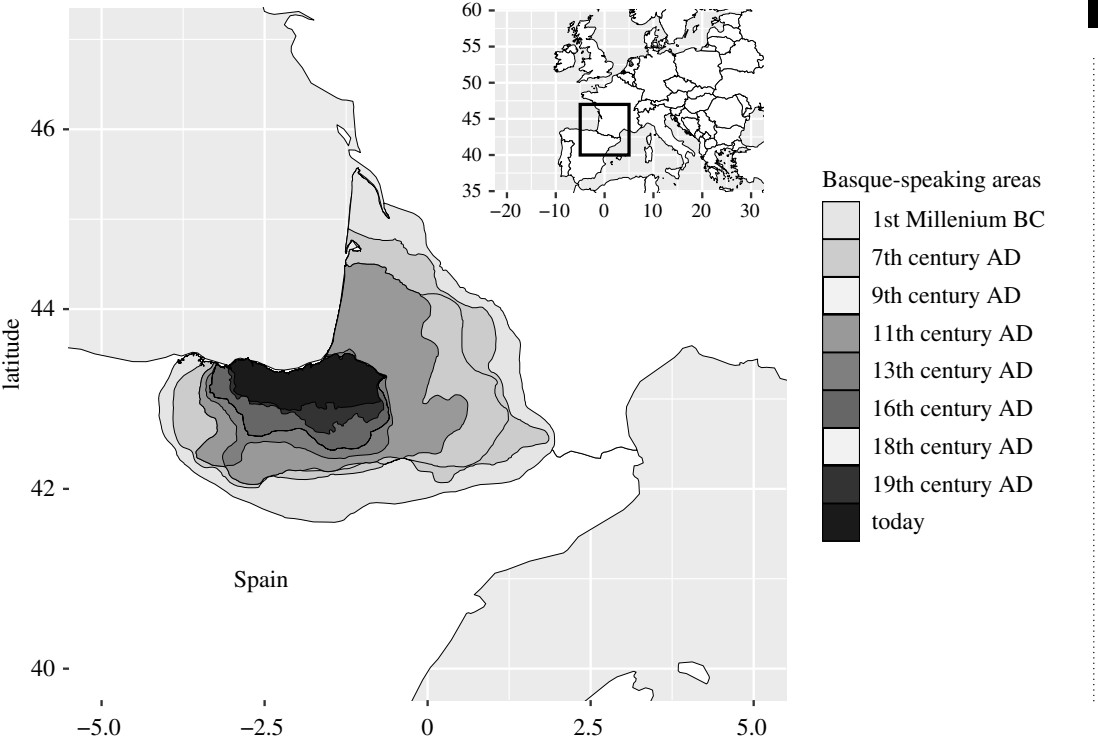

**Figure 2.** Historical reduction of the Basque-speaking areas. Adapted from https://commons.wikimedia.org/wiki/File:Euskararen_atzerakada1.svg and https://commons.wikimedia.org/wiki/File:Euskararen_atzerakada1.png, in turn, based on [28–31].

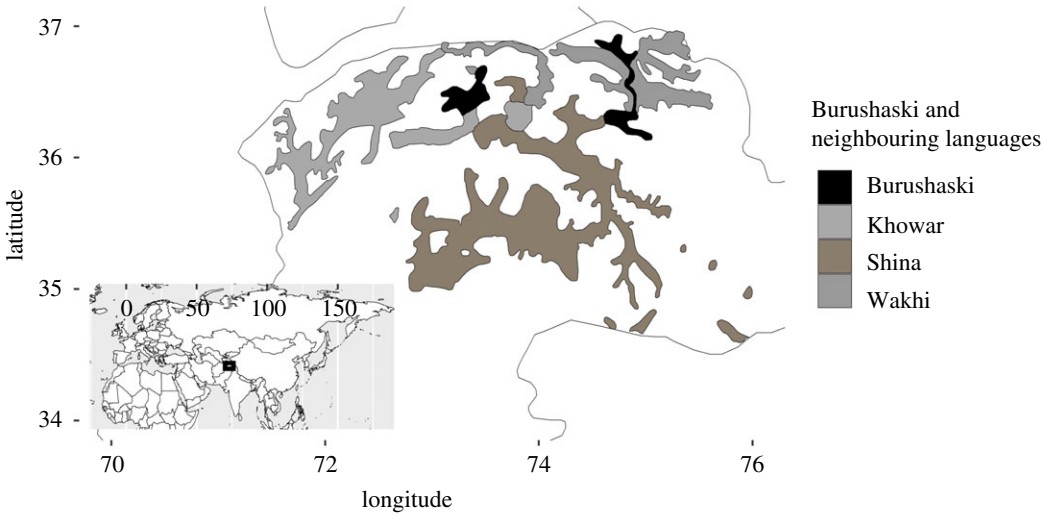

**Figure 3.** Burushaski and neighbouring languages in northern Pakistan. Based on [33].

Burushaski was once spoken in a wider area [34,35], in particular, further south [36], though the maximal distribution is not well established [37]. Perhaps its use extended to all of Dardistan [38]. On the lower courses of the Hunza river, the Indo-European Shina language is spoken. This essentially bisects the Burushaski-speaking area to the upper courses of the Hunza and Yasib (figure 3), where the language is spoken in slightly different varieties [34]. This distribution alone is suggestive of a scenario in which Shina replaced Burushaski along the lower courses of the rivers [39], and indeed this is also how language contact effects and cultural patterns are interpreted [35,40]. Consistent with this scenario, the Burusho people—the speakers of Burushaski—have repeatedly been considered a relic population whose presence in the area predates the arrival of Indo-European speaking people [34,35,38,41]. Parallel to the Basque case, '[t]he country with its physical savageness and poverty could never have attracted the ambition or cupidity of any sane conqueror, and with easier routes available to the West there is little reason to believe that any serious invader has traversed or even

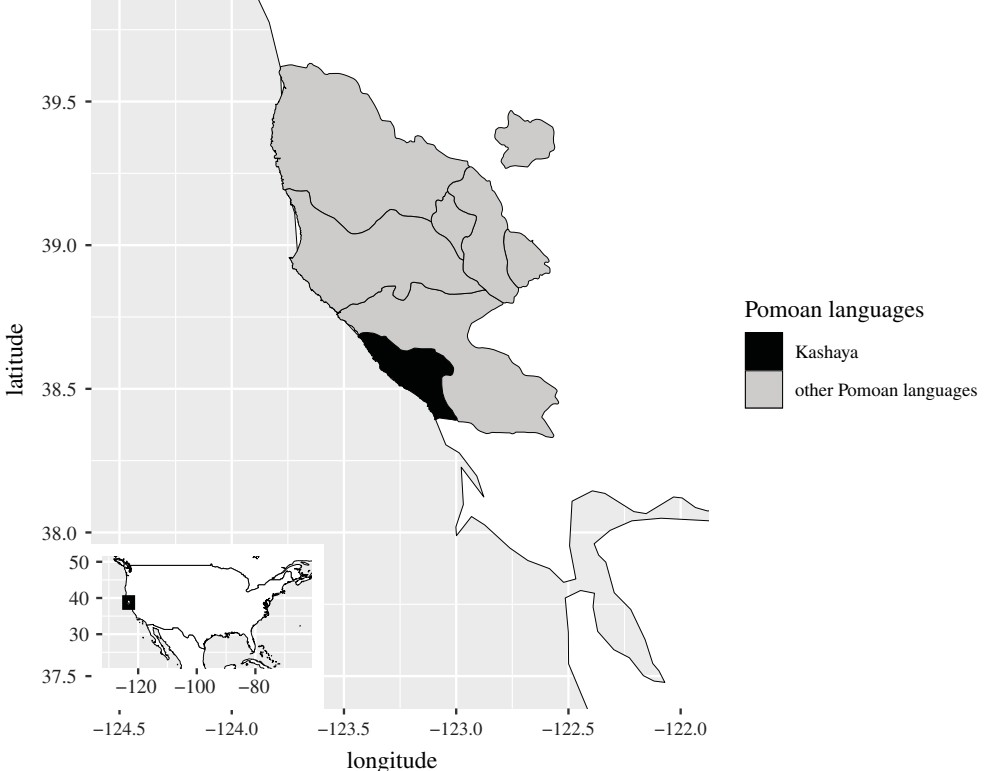

**Figure 4.** Original distribution of Pomoan languages, adapted from [42].

penetrated it in force' [34, p. xxxiii]. In other words, Burushaski has remained at the highest and most difficult to access areas of an originally wider distribution.

We also see similar patterns in the reduction of language families and their branches. Pomoan (figure 4) is one of the many low-level language families of California, originally consisting of seven relatively closely related languages (Pomoan is typically included in the proposed 'Hokan' macrofamily, but in spite of intense and serious historical-comparative research [43,44] the proposal has so far not yet been demonstrated with the level of certainty that would satisfy all).

As is typical for hunter–gatherers of California and elsewhere, the original populations of Pomo groups, and accordingly the number of speakers per language, were small. For the Kashaya, one of the smallest Pomo groups originally [45], the original population has been estimated as comprising 550 [46] or between 800 and 1200 [47] individuals. Today, however, Kashaya is the Pomoan language with the largest remaining speaker population. Kashaya has thus moved from originally one of the smallest Pomoan languages to the most vigorously spoken language, and is likely to be the only surviving language soon, i.e. one that would, without historical information, be considered an isolate. '[W]hy a Pomo division that was formerly among the smallest has survived as the largest culturally distinct group' [45, p. 3] is of interest for present purposes. In part, it is owing to the vicissitudes of history (the Kashaya, unlike other Californian Indians, made the first contact with Russians who did not resettle them or force them into labour and treated them relatively well [45]). However, the historical fate of the Kashaya and the patterns of contact with whites is also predicated on the physical environment: agricultural lands were comparably poor in the Kashaya-speaking areas and access to the coast was difficult, making the land unattractive for the foundation of major European settlements oriented towards the exploitation of coastal resources [45]. In other words, the lands exploited by the Kashaya were of marginal interest to nineteenth century settler society [46].

Similar processes can be observed at still larger geographical and temporal scales, prominently in the history of the Celtic branch of the Indo-European languages. The geographical and cultural context of the origins of Celtic continue to be debated [48–51]. The agreement on the maximal extension of Celtic speech is also not perfect, but the picture here is less contentious: Celtic was spoken in a broad belt across continental Europe from the Atlantic coast to the Black Sea, on the British Isles, significant parts of the Iberian peninsula, and, as a result of long-distance migration, a small area of Anatolia (figure 5).

Today, Celtic languages—Breton, Irish, Scottish Gaelic and Welsh—remain in areas on the westernmost coastal fringes of Europe and the British Isles (figure 5), having been superseded in large

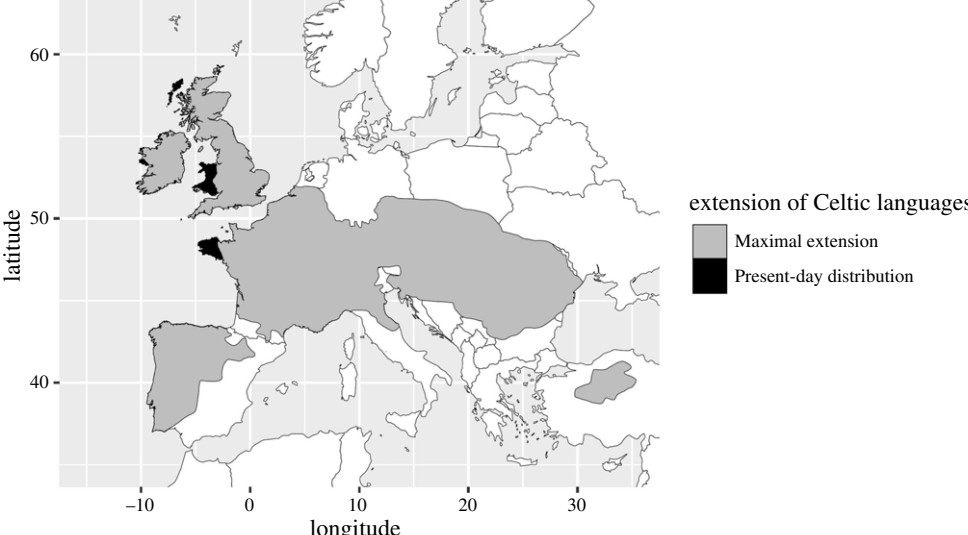

**Figure 5.** Reconstructed maximal extension of Celtic linguistic evidence from [52], 'reassuringly in agreement' with independently arrived at accounts [53] and present-day distribution from [54–57]. Where sources differentiate percentages of the population for which a Celtic language is native for different regions, the figure shows areas where the majority of the population retains knowledge of the Celtic language.

parts of their former distribution by later language spreads (principally Germanic and Romance). The retreat of Celtic in continental Europe was temporally and regionally uneven [58], though Celtic speech seems to have given way to Latin earlier in northern Italy than in other parts of Gaul. The Celtic languages are a branch of a family, not an isolate. Nevertheless, their history shows that similar processes to those that have led to the reduction of isolate's ranges are operative not only in small families like Pomoan, but also within large families on comparably large scales. Indeed, had the reduction of the Celtic range to the fringes of Europe taken place a few millennia earlier, the genealogical signal linking them to Indo-European would probably have been lost.

The anecdotal qualitative evidence in the preceding section can serve to inform a model for isolate genesis that can be tested empirically on continent-wide scales. A major quantifiable factor seems to be the proximity to major geographical barriers, most prominently the coastline and mountain areas with conditions that make them unsuitable for permanent habitation. Generally, Basque, Kashaya and Burushaski have shown that the limited productivity of rugged terrain has prevented, or delayed and mitigated, major language spreads, highlighting one aspect of mountain topography for linguistic geography (see a more detailed qualitative survey of this in [11] and quantitative modelling in [7]).

Several parallels have been drawn between language development and biological evolution, ontologically as well as methodologically [59,60]. It is, therefore, worth digressing briefly to ask if there is any analogous biological concept to the processes sketched here for the historical dynamics of language ranges and isolate genesis. It is tempting, not least because of the etymological connection between the linguistic terminology and the prevailing terminology in population genetics, to relate the possible pathway towards language isolates via language reduction towards geophysical barriers to the notions isolation by colonization, isolation by distance and/or isolation by environment. The former two have recently been adopted to describe spatially structured linguistic variation in Japan [13]. The processes at stake here seem to resemble more closely isolation by environment patterns, 'in which genetic differentiation increases with environmental differences, independent of geographical distance' [61, p. 5650]; this is qualitatively similar to the observation that marginal ecological domains in which isolates, according to the emerging model, are not reached, or reached only late, by later language spreads. One of the mechanisms that brings about isolation by the environment is selection against immigrants [61]; this is comparable to the role that agriculture-based subsistence (arguably an adaptation) and related preferences for easily arable land has played in making lands where isolates remain unattractive, as in the case of Basque. The role of physiogeographical factors furthermore dovetails with the notion of refugial isolation, where a large number of endemic species evolve in characteristic environments involving mountainous topography and geographical barriers [62].

While the cases surveyed provide only anecdotal evidence, salient aspects of the present-day distribution of isolates as plotted in figure 1 are consistent with a model in which language isolates

are considered the result of qualitatively similar reductive processes. One relevant observation is that unrelated languages spoken by non-agriculturalist people survived the spread of agriculturalist Uto-Aztecan people on the coast of northwest Mexico [63] (consistent with existing theorizing, isolates and more generally 'survivor' languages indeed often occur on the edges of major spreads [15,64,65]). However, the relevance of the parameters appears to be broader, and they also appear to play out in larger areas with less clear scenarios as to the large-scale diachronic language dynamics: California is a hotspot of isolates, and in the linguistically less diverse eastern half of North America, there is a pearl chain of languages considered isolates on the Gulf coast (though a genealogical relationship of these is possible [66]). Also in Australia, isolates cluster on the north coast, and in Papua New Guinea, a generally highly diverse linguistic microcosm, isolates are more frequent in the lowlands between the northern coast and the New Guinea highlands than in the highlands themselves, where a large ancient language family, Trans New Guinea, dominates (though this language geography is also amenable to other interpretations [67]). On islands, we would expect the same pattern. In fact, however, isolates are rarely spoken on islands. Rather, newly arriving languages on islands tend to sweep through [68], ousting any previous diversity that may have existed. In fact, if it is major language spreads in relation to physical boundaries which drives the genesis of language isolates, these dynamics may only play out on continental scales.

The two perspectives on isolates and models for their genesis both relate to the dichotomy of linguistic spread zones versus residual zones (later called accretion zones) [1,2]. Accretion zones are areas characterized by high levels of linguistic diversity and typically deep families; '[l]anguage isolates and isolate families are likely to be found in residual zones' [1, p. 21]. Consistent with the cases presented here, such accretion zones tend to occur on the periphery of spread zones, and they also tend to preserve a certain typological profile over significant periods of time [1]. However, there are potential difficulties in determining the boundaries of spread zones and residual zones, and at times conventional, more or less arbitrary boundary lines have been used [1]. This lack of clarity on what can count and what cannot count as a spread or a residual zone is one of the main reasons of criticism of the distinction [69]. Evaluating the two models for the genesis of isolates developed above statistically thus, on the methodological level, contributes to putting the notions of spread and residual zones, widely referred to in linguistics for their descriptive usefulness, on more solid quantitative footing.

# 3. Analysis and results

The two models for the genesis of isolates make different predictions that are amenable to statistical analysis: if isolates are a manifestation of general linguistic diversity, there should be a significant association between isolate location and general levels of linguistic diversity. If the alternative account that emphasizes reductive processes and geophysical factors, on the other hand, is generalizable and contributes significantly to the global distribution of language isolates, this should not be a strongly recognizable pattern, and one would rather expect isolates on major landmasses to occur more frequently close to major geophysical barriers, prominently coastlines and uninhabitable mountain areas, than non-isolates.

To assess these predictions, geolocations of languages from Glottolog 4.2 [19] were used, and isolate status was also derived from Glottolog 4.2. Distances to mountain areas on the same landmass were calculated on the basis of the Global Mountain Biodiversity Assessment (GMBA) mountain inventory dataset 1.2. [70]. Given that the qualitative survey indicated that mountains are especially relevant to shaping language geography if they represent major geographical barriers to population movements, only mountain areas with alpine zones that are not permanently inhabitable by humans were considered for the main analysis (belt 1 in [70]). Languages not spoken on a landmass on which such a mountain area is located were removed from the dataset for the main analysis; this decision receives independent empirical support from the observation that isolates tend to not occur on (small) islands [68]. However, ancillary analyses take into account landmasses with any mountain area recognized in [70], not just those which feature alpine conditions. In addition, in further ancillary analyses languages which Glottolog 4.2 lists as 'unclassified' were included among the language isolates. See data and methods for more details. Unless otherwise indicated, results reported here come from the main analysis that considers languages on landmasses with alpine mountain areas ($n = 5251$) and that is based on isolates as recognized in Glottolog in a narrow sense (i.e. excluding 'unclassified' language) ($n = 121$).

Descriptively, the median distance of isolates to the nearest coastline is approximately 309 km (mean 409.0604 km, s.d. 368 km) and 489 km (mean 855 km, s.d. 1009 km) from the nearest alpine mountain

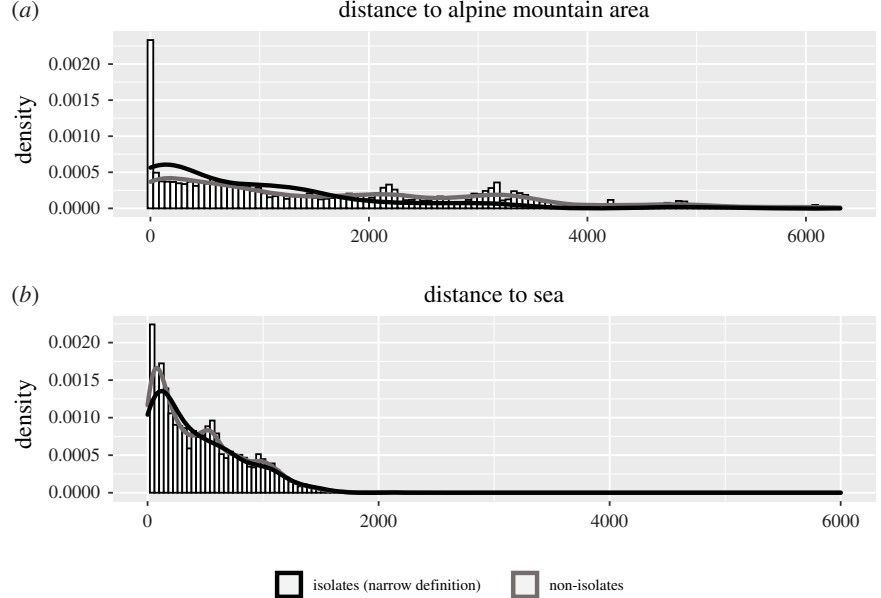

**Figure 6.** Density plots of the distance of isolates and non-isolates to the sea (*a*) and high mountain areas (*b*).

area on the same landmass, whereas the median distance of non-isolates was greater from both coastline at 346 km (mean 442 km, s.d. 383 km) and alpine mountain area at 1255 km (mean 1654 km, s.d. 1488 km) (see the electronic supplementary material, table S1 for values on the dataset taking into account all mountain areas). The distribution exhibits a strong left skew, with many more languages found very close to the coastline and to high mountain areas (or in them) than far away from them. Figure 6 shows histograms of the distance to alpine mountain areas and coastlines for the entire dataset, overlaid with density plots of the distributions while distinguishing the densities of isolates and non-isolates.

These statistics suggest that indeed the density of isolates is higher in proximity to or in high mountain areas than non-isolates, whereas for distance to the sea, the distribution of isolates is less smooth, but overlaps with that of non-isolates.

As a first formal test of the situation, given the strong skew of the variables I employ the non-parametric Mann–Whitney–Wilcoxon test, treating isolates and non-isolates as distinct populations and testing the alternative hypothesis that the distance to the sea and alpine mountain area is lower for isolates than for non-isolates. As suggested by the larger difference in the mean values between the two groups, the test supported the alternative hypothesis in the case of distance to mountains ($W = 211644$, $p < 0.000001$), but not in the case of distance to the sea ($W = 295520$, $p \approx 0.18$). Results of ancillary analyses when including 'unclassifiable' languages among the isolates showed a trend for such isolates in a wide sense to be closer to alpine mountain areas, but did not reach conventional significance thresholds (see the electronic supplementary material, table S2).

This result is a first indication that there indeed may be a significant difference between isolates and non-isolates with regard to the distance to major geographical barriers. However, depending on how isolates are defined and what mountain areas are considered, in particular, the distance to the coast as a relevant variable is questionable.

To explore this further and to assess the robustness of this first result, I resort to Bayesian mixed effects modelling. This is called for because an exploration of the data structure shows that the distance to alpine mountain areas differs saliently between different continents. In Africa, the only mountain area with alpine conditions as recognized in [70] is Mount Kilimanjaro in southeastern Africa, yielding particularly large distances for instance for supra-Saharan languages. Figure 7 shows the global distributions as well as the distributions for Africa, Eurasia, North America, 'Papunesia' and South America separately.

Bayesian mixed effects logistic regression models were run in R using the package brms [71]. The models attempt to predict the distance of language isolates from mountain areas with alpine zones, distance to the coastline and the interaction of distance to alpine mountain areas and the coastline, while controlling for possible differences between major geographical regions of the world using a

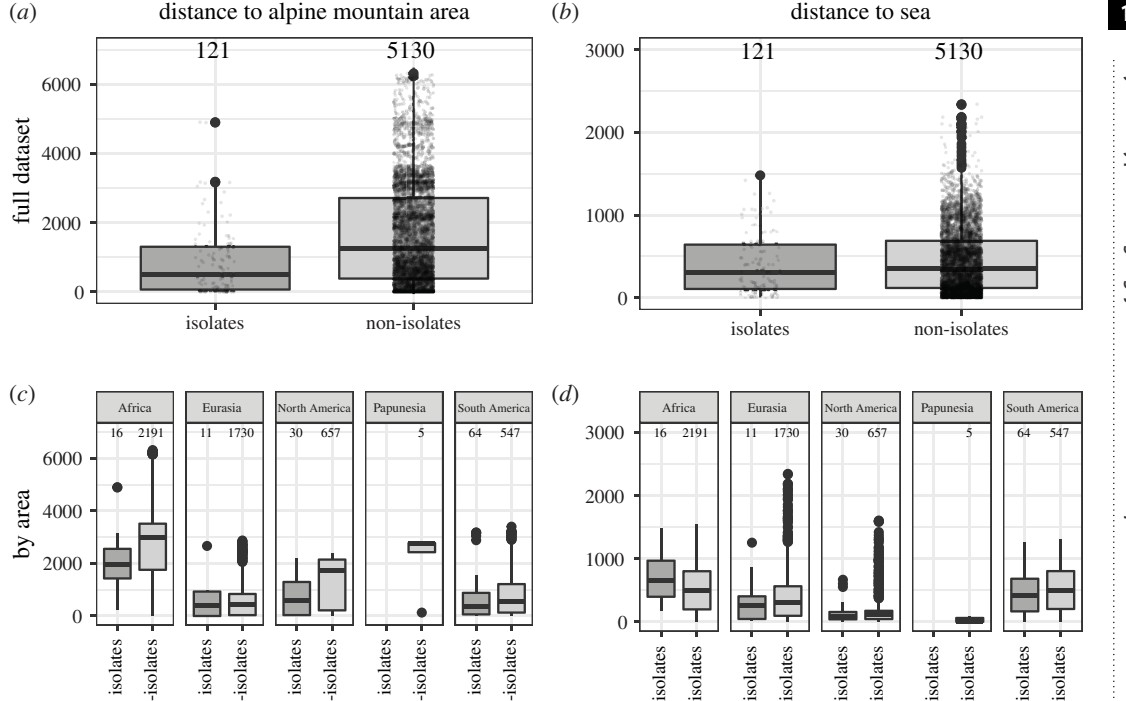

**Figure 7.** Boxplots for distance to alpine mountain area and sea for isolates and non-isolates. (*a*) Distance to alpine mountain area for the full dataset; (*b*) distance to the sea for the full dataset; (*c*) distance to alpine mountain broken down by macro-area; and (*d*) distance to sea broken down by macro-area.

breakdown of the world into six macro-regions (Africa, Eurasia, Papunesia, Australia, North America and South America) that is implemented in Glottolog 4.2. Australia is irrelevant in the present context for not hosting any isolates, and Papunesia was not included in the random effects structure because of the small number of remaining observations (with negligible consequences on the result, see data and methods section). The result for the main analysis is that increasing distance to the sea by a factor of 10 increases the log odds of observing an isolate by 0.01 with a 95% credible interval (CI) of [−0.84 1.19], and distance to a mountain area with alpine environments decreases the log odds of observing an isolate by −0.13 (95% CI [−0.75 0.49]), respectively. This means that neither distance to the sea nor distance to alpine mountain areas has a credible effect on the distribution of isolates. Neither has the interaction between distance to sea and distance to uninhabitable mountain areas such an effect (estimate 0.04, 95% CI [−0.19 0.44]). The predictive accuracy of the model was approximately 76%, and the predictive accuracy of isolate status was slightly higher even at approximately 77%. Similar results were obtained when assessing isolates in a wide sense and considering all mountain areas rather than just ones with alpine belts (see the electronic supplementary material, table S2).

In sum, there was no support from Bayesian mixed effects logistic regression for the idea that the present-day locations of isolates generally reflect processes of language reduction as derived from the case studies. Similar results were obtained for ancillary analyses (see the electronic supplementary material, table S2).

Turning to the alternative view that views language isolates as one particular manifestation of high linguistic diversity generally, I looked at the association between isolate location and general levels of linguistic diversity on the language level ('language richness' in terms of [6]). I have tested the spatial association between the distribution of isolates and non-isolates using a Spatial Point Pattern Test [72] as implemented in the R package sppt [73]. This test is designed to test the similarity between two spatial point patterns—here, the distributions of isolates and non-isolates—in predefined areal units— here, landmasses—and derives a global statistic, $S$, for the similarity between the two patterns. Testing differences between the proportions of isolates and non-isolates on landmasses on which language(s) are spoken, the association was strong ($S = 0.75$) (the association became even stronger when considering landmasses with mountainous terrain generally and the full dataset; electronic supplementary material, table S2).

# 4. Discussion

In this article, I have discussed two alternative points of view on language isolates that both follow naturally from their status as self-contained genealogical units and observations on their worldwide distribution: one which views isolates as a specific manifestation of the more general phenomenon of linguistic diversity, and another, elaborated on the basis of case studies on the history of known isolates and other processes in which genealogical groupings were reduced in their range, that emphasizes the diachronic dynamics of language reduction and the role of geophysical barriers.

Statistical analysis showed a strong association between isolates and overall language richness. This would suggest that whatever drives general language diversification also drives the distribution of isolates, as seems to have happened in the Upper Amazon. This underscores that isolates are not in principle different from other languages [17]—in fact, they may exhibit internal diversity (as do e.g. Basque and Burushaski) that are the potential seeds of future small language families.

The alternative hypothesis received no systematic support from statistical analysis. On a worldwide level, without taking into account possible differences between different parts of the world, statistical testing did suggest a significant effect of distance to major alpine mountain areas, but not distance to the sea, on the distribution of language isolates. Although the results of Bayesian mixed effects modelling likewise suggested that distance to alpine mountain areas was the more relevant variable than the distance to the sea, the effects of both variables were, in fact, slight and not credible. It is plausible to assume, therefore, that the global results arise out of the influence of particular areas of the world, but are not generally applicable. When taking into account differences between different parts of the world, there was little evidence for major geographical barriers influencing the distribution of isolates as would be suggested by the case studies.

Both the conceptual models and the statistical analyses used here to explore their merits are basic and best thought of as starting points for further exploration. There are many ways in which the models and the statistical analyses could be refined and developed further. Conceptually, the processes that drive language diversification, and, if conceived of in this way, also isolate genesis, are poorly understood. Here, I have only offered some observations on the Upper Amazon and the potential role of the Marañón as a vector that, in the long run, may have played a role in creating a linguistic landscape featuring both high general language diversity and a high number of isolates. On a worldwide level, the emergence of the highly uneven patterns of linguistic diversity, and in particular the social and linguistic processes behind these and their relationship to the environment, is a current focus of research and debate [6–14], and advances will probably also feed into a better understanding of the when, why and how of isolate genesis. Regarding the alternative hypothesis, which received no support in the analyses, operationalization of the processes of language range reduction that were observed in the case studies through distance to the coastline and uninhabitable mountain areas may be too simplistic. For instance, the qualitative case studies showed that terrain rugosity and limited agricultural possibilities, which are not taken into account in the present simple model, are relevant in mitigating or entirely preventing the influx of expansive populations that would ultimately lead to language replacement. Indeed, rugosity has been identified in at least one study as a relevant factor in driving language diversity [7], although this result could not be replicated [12]. Taking into account ecological parameters more systematically, disentangled from the distance to physiogeographical barriers that are used to operationalize the isolatogenetic model based on reductive processes here, seems advisable; sociolinguistic factors might be taken into account as well.

This leads to possible improvements in analytic approaches, in line with a robustness framework in research on language diversity that emphasizes incremental research and multi-pronged analyses [74]. Consistent with the possibility that more complex models are needed, assessments of the robustness of the result could also involve the use of different, polygon-based datasets of language ranges. These would allow the elimination of uncertainties and imprecisions in calculating distances to geophysical barriers on the basis of abstract point coordinates, which are especially virulent for languages that are spoken over wide areas. These are not yet available widely in high-quality, but efforts in this direction are increasing [75]. Also, different statistical approaches and techniques might lead to additional insights. As one reviewer suggests, one might look at regions of the world with geographical and environmental properties that are of potential interest for language diversity (such as alpine mountain areas) and well-defined surrounding areas and look at the number of isolates and non-isolates within and outside of the search spaces thus defined (as is done for instance in [76]). In a different vein, sophistication could be increased by space–time regressions that are informed by direct historical

evidence on language reduction as well as geophysical information. Finally, it is important to realize that there is no sharp qualitative difference between a dialectally diverse isolate and a small language family, making it potentially revealing to expand the scope of the investigation to small families, and/or, as suggested by a reviewer, to move away from the simple binary distinction between isolate and non-isolate towards variables such as the number of languages per family or branch, with isolates then representing the extreme end on a continuous scale rather than being modelled as a distinct 'type' of language.

More generally, it may be simplistic to think of the two alternative accounts of isolate distribution—one that views them as a result of general mechanisms of language diversification and the other as driven by gradual isolation in proximity to major geographical barriers—as clearly distinguishable alternatives. To begin with, language change can happen through textbook processes of sound change and gradual lexical replacement that occur in all languages and eventually make the genealogical signal too faint to recognize. But there is possible sociolinguistic influence on communicative behaviour—as when e.g. Australian languages are said to reach a 'cognate equilibrium' through massive conscious avoidance and replacement of inherited lexical items for cultural reasons [77], cf. [78,79]. With a view specifically to environmental factors, it has been realized that drivers of diversity may be different in different parts of the world, with the overall picture emerging as the result of several different processes [14]. In fact, under more sophisticated modelling it may turn out that neither of the two models evaluated here is wrong, but that the present-day location of isolates is the result of different processes, with some isolates (perhaps a limited number) being the result of language reductions near major geophysical barriers in a manner that is consistent with the case studies of §2, and other isolates the product of the still poorly understood social processes that drive language diversity on local and global scales. Indeed, the conflicting results which many studies of this diversity have yielded so far suggest that a complex web of factors probably underlies present-day patterns of worldwide language diversity.

# 5. Data and methods

## 5.1. Data

### 5.1.1. Language data

Linguistic data analyzed in this article come from the Glottolog database v. 4.2 [19]. Glottolog furnishes a catalogue of the world's languages with genealogical classification as well as point coordinates to represent the approximate location where they are spoken. This is often unproblematic for local languages with a limited number of speakers and a restricted area in which they are (or were traditionally) in use, as is the case for many of the minority languages that make up the bulk of the world's linguistic diversity. It is more problematic in the case of 'major' languages, typically national languages, which are spoken over wide areas or even on several continents, and which have relatively many speakers. Policy in these cases seems to have been to choose a coordinate that reflects a historical centre of distribution. For instance, Spanish is placed on the Iberian peninsula and Russian appears to have been placed at or near the latitude and longitude of Moscow. In spite of this disadvantage compared to alternative polygon-based representations of a language's distribution, point coordinates continue to be used widely in quantitative studies of linguistic diversity (e.g. [9]) for the analytic and computational advantages they offer. Nevertheless, they are, whether in the case of local or widely distributed languages, a significant abstraction from a language's actual range.

For analysis, I have excluded from the dataset languages which Glottolog only keeps for 'bookkeeping' purposes (which are assigned to a pseudofamily with ID 'book1242'). I have also excluded artificial languages (pseudofamily ID 'arti1236'), and, after some deliberation, did the same for sign languages (pseudofamily ID 'sign1238'). While, in spite of the modal differences, the development of sign languages is subject to processes that are remarkably similar to spoken languages (including e.g. grammaticalization and dialect development), it is not clear to what extent they behave similar to spoken languages as regards to the spatial dimension of diversification and change [80], and there is no clear analogue to the notion of the higher-language family for sign language [81] (though cf. [82]). All other data were retained for analysis (including mixed languages and pidgins and creoles, which are assigned to pseudofamilies in Glottolog), leaving a total of 7989 languages.

Isolates in Glottolog can be identified easily because, unlike languages belonging to language families, they are not assigned a family ID. However, there is a pseudofamily of 'Unclassifiable' languages (pseudofamily ID 'uncl1493'). This is a difficult category. On the one hand, it includes extinct languages whose existence is mentioned in historical documents, but for which there is no primary data available whatsoever. Such languages are indeed unclassifiable. On the other hand, Glottolog's 'Unclassifiable' pseudofamily also includes languages of which we have some knowledge, but where the documentation is insufficient (owing to limitations in terms of quantity and quality) for a consensus genealogical affiliation to be reached. The boundary between such 'unclassified' languages to isolates is not clear-cut, and the assessment is to some extent subjective. I have, therefore, created one category of languages that includes only isolates in a narrow sense (excluding 'unclassifiable' languages), and one category of isolates in a wide sense, which includes isolates in the narrow sense as well as 'unclassifiable' languages. Because the latter category is problematic for several reasons, main analyses focus on isolates in a narrow sense, but all analyses are also carried out for the larger group, and results are reported in the electronic supplementary material, table S2.

### 5.1.2. Geographical data

For analyses, precise definitions of what constitutes a mountain area are necessary. However, such areas are surprisingly difficult to define objectively [83,84], and subjective perceptions are often locally relevant [83]. Here, I use the GMBA mountain inventory dataset v. 1.2. [70], which provides a standardized inventory of 1048 mountain areas and associated polygon shapefiles. V. 1.1. of the dataset also defines climatic belts. Depending on latitude, life conditions at similar elevations may be drastically different so that elevation alone would not allow meaningful comparison of climatic and environmental conditions in different mountain areas. Because the inventories of v. 1.1. and the current v. 1.2 do not overlap precisely, automatic joining was not possible. Instead, I have manually created a subset of mountain areas in the v. 1.2 dataset in which the alpine environments which feature conditions that prevent permanent inhabitation by humans are present. Figure 8*a* shows the mountain areas distinguished in the GMBA dataset v. 1.2., with mountain areas in which alpine conditions occur at the highest elevations as per v. 1.1 highlighted. Because the model for the emergence of isolates requires that mountains constitute a significant physical obstacle for the spread of populations and their languages, as e.g. in the case of Burushaski, and its applicability is restricted to continent-sized areas in which such mountain areas occur, I have consecutively removed from the Glottolog dataset those languages spoken on landmasses which lack mountain areas as defined in the GMBA inventory dataset v. 1.2 (figure 8*b*), leaving 7117 languages, and then also languages spoken on landmasses in which mountains with the highest (alpine) climatic belt absent (figure 8*c*), leaving 5251 languages. The OpenStreetMap land polygon dataset (https://osmdata.openstreetmap.de/data/land-polygons. html) was used to represent the world's landmasses for this purpose. This last dataset of 5251 languages was used for the main analysis. For each language in this dataset, the distance to the nearest mountain area with alpine regions on the same landmass was identified, and then the distance to that mountain area was calculated using the st_distance function of the R package sf [85,86]. For the more extensive dataset that takes into account all mountain areas regardless of whether they feature alpine conditions, the distance to the nearest mountain area on the same landmass was calculated in the same manner.

Distance to the coastline for the languages in the datasets was calculated applying the st_distance function to a sf object created by the ne_coastline function of the R package rnaturalearth [87].

# 6. Methods

## 6.1. Wilcoxon–Mann–Whitney test

For the first global assessment of the difference between isolates and non-isolate languages regarding distance to the sea and alpine mountain areas, I have applied Wilcoxon–Mann–Whitney tests [88,89] as implemented in the function wilcox.test from base R. Such a non-parametric test, which does not require data to be normally distributed, is required for analysis of the present dataset with strong left skews in the key variables.

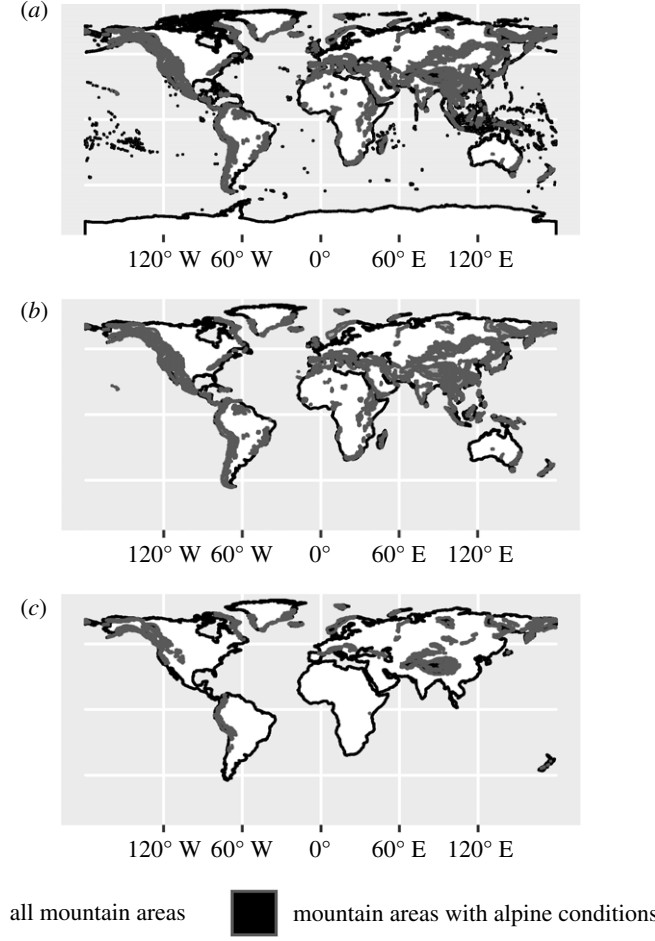

Figure 8. Illustration of subsetting landmasses depending on the presence or absence of mountain areas as defined in [70]. (*a*) All landmasses in the dataset and mountain areas; (*b*) excluding landmasses without any mountain area; and (*c*) excluding landmasses without any mountain area of the alpine category.

## 6.2. Bayesian logistic mixed effects regression

To test the idea that the distribution of isolates is related to major geophysical barriers in a regression framework, Bayesian logistic mixed effects regression models were run using RStan [90] as accessed through the R package brms [71]. I applied logistic regression because the dependent variable is binary: a language is either coded as an isolate (either in a narrow sense for the main analyses or in a wide sense for ancillary analyses) or as a non-isolate. I applied mixed effects regression because it is standard in the analysis of typological data to control for possible differences between different continent-sized parts of the world, so-called macro-areas. This is particularly important here, since, as figure 7 shows, there are indeed significant differences between measures in such macro-areas as defined in [91] and as implemented in Glottolog 4.2. [19]. Included fixed effects were the distance to the coast and distance to mountain areas in kilometres. The fixed effects were subject to log10-transformation to reduce skew after a constant of 0.00001 km was added to all datapoints; this commonly applied practice is not neutral, yet better practices that estimate the optimal constant to be added are not yet implemented in R [92]. Models also included an interaction term to assess the possibility that it is proximity to shorelines together with proximity to mountain areas (as e.g. in California) that accounts for the distribution of isolates. A random effect for macro-area completes the model structure. Although the median distance to sea and mountain areas differ widely by family, I have not fitted a random effect for genealogical control as isolates by definition do not belong to a larger language family, and therefore the random effect structure would be correlated with the response. Mountain areas classified as 'alpine' in [70] for Glottolog's 'Papunesia' macro-area occur only in New Zealand. New Zealand hosts just five languages in the Glottolog dataset, none of which is an isolate; I have, therefore, decided to remove the Papunesia area prior to analysis, and leave the

macro-area for New Zealand languages uncoded. The main model was run for isolates as narrowly defined in Glottolog (i.e. excluding 'unclassifiable' languages) and the distance to mountain areas with an alpine belt; ancillary models were constructed with 'unclassifiable' languages treated as isolates, and, at the request of a reviewer, a further model that retains the 'Papunesia' macro-area was run to assess the impact of the decision to remove this level from the random effects structure. I have placed a weakly informative prior of s.d. = 2 on fixed effects, a flat prior on the interaction term, and have otherwise used default priors (Student's $t$ with three degrees of freedom, prior location of 0, and a prior scale of 2.5) for the intercept, standard deviation of random effects and residual errors. Each model was run in four chains with 3000 iterations, with the drift parameter delta set to 0.999999 and a maximum tree depth of 20 to avoid divergent transitions; 2000 of the 3000 iterations in each chain were used for warm-up. $\hat{R}$ values of 1 for each parameter, effective sample size estimates and a visual inspection of the chains indicated that all models converged, and comparisons of plots of observed data with posterior predictive samples showed that the models fit the data well. The ancillary analysis retaining the 'Papunesia' macro-area showed that the results are affected only very slightly by the decision to remove it from the random effects structure (see the electronic supplementary material, table S2 for full information). The vast amount of languages of the world belong to families and are not isolates. Hence, the number of non-isolates in the dataset greatly exceeds those of isolates, i.e. statistically speaking it is zero-inflated. Zero-inflated Bernoulli models [93] are not implemented in brms, and their application would also be problematic for theoretical reasons because they assume that the observed zeros are generated by two different processes. While this makes sense in many contexts, here it would be unclear in particular what the real-world process of language development that only ever creates isolates would look like. To assess the reliability of the models given the resulting danger of zero-inflation and overdispersion post hoc, instead, the predictive accuracy of the models was assessed following §11.2.4 of [94].

## 6.3. Spatial Point Pattern Test for the geographical association between isolates and language diversity

To explore the geographical relationships between isolates and non-isolates, a statistic that assesses the similarity between two spatial point patterns is needed. Here, I have relied on the Spatial Point Pattern Test as implemented in the function sppt_diff() in the R package sppt [73]. Originally developed for application in criminology, this is an area-based test that directly tests differences in proportions of observations in the areas. For present purposes, these are those landmasses in the OpenStreetMap land polygon dataset (https://osmdata.openstreetmap.de/data/land-polygons.html) on which at least one language is spoken as per Glottolog 4.2. As used here, the test assesses differences in proportions based on Fisher's exact test (which is conservative and applicable also when the number of observations is small), with an overall test statistic being based on results that are automatically corrected by the sppt_diff() function for multiple comparisons.

Data accessibility. Data and relevant code for this research work are stored in GitHub: https://github.com/urban-m/isolates and have been archived within the Zenodo repository https://doi.org/10.5281/zenodo.4621315. An R workspace and additional datasets are available via the Zenodo repository https://doi.org/10.5281/zenodo.4621263.
Competing interests. I declare I have no competing interests.
Funding. This research was supported by the German Research Foundation (DFG—Deutsche Forschungsgemeinschaft), Grant no: UR 310/1-1 and publication was supported by the University of Tübingen's Open Access Publishing Fund.
Acknowledgements. I am grateful to Gerhard Jäger and Christian Bentz for commenting on earlier versions of this article. I am also indebted to two anonymous reviewers for their detailed and thoughtful comments and suggestions that helped improve the text significantly. The limitations of this article and any errors of fact or interpretation it may contain should be attributed to me, not any of the mentioned colleagues.

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
