## [Peer Review File · Royal Society Open Science]

Review History

RSOS-202232.R0 (Original submission)

Review form: Reviewer 1

Is the manuscript scientifically sound in its present form?

Yes

Are the interpretations and conclusions justified by the results?

Yes

Is the language acceptable?

Yes

Do you have any ethical concerns with this paper?

No

Have you any concerns about statistical analyses in this paper?

No

Recommendation?

Accept with minor revision (please list in comments)

Comments to the Author(s)

The current paper is mostly clearly written (there are a few garden path sentences and typos, and see below on the introduction) and makes a clear and interesting contribution on the geographical distribution of linguistic isolates (not sure it does about 'evolution', though).

This paper has a lot of potential, but suffers slightly from structural issues detailed below. In addition, the analyses it presents are basic, and given the alternatives listed by the author, and the alternatives one thinks of while reading, it feels somehow disappointing that the author hasn't taken the extra steps to do follow through with one or more of these options. However, the work reported on here has to be done before doing more elaborated analyses. I really appreciate the case studies reported on in section 3 and how they lead to hypothesis formation.

Structural issues. I had to read the introduction several times to identify the two opposing views, summarized by:

1: [Isolates] "are generated through the same general historical processes that drives language diversification. As such, it should be the case that linguistic isolates occur in hotspots of linguistic diversity."

2: [Isolates are] "the last surviving member of a former language family [16, 18]. This stance is supported empirically by languages which currently qualify as language isolates, but which are known to have had related members", i.e. we will find isolates in places where they are pushed to following subsequent language spread, "towards a major geographical barrier, prominently the coastline or a major mountain area with conditions that make it not permanently habitable"

Even this summary is somehow not as clear as it could be. The main hypotheses on p. 9, "f the account based on the qualitative survey ..." are much clearer.

Conclusion/Spatial Point Pattern Test.

The author concludes (p. 11) that the first hypothesis listed above is right. However, a lot of time is then spend on explaining how hypothesis 2 could still be true. The two hypotheses are not given the same magnitude throughout the paper. Hence, the Spatial Point Pattern Test is not explained at all, nor are alternatives to investigating the relationship between "isolate location and general levels of linguistic diversity on the language level ("language richness". Is the Spatial Point Pattern Test the best possible test? Nor is there much depth given to the processes that would lead to more isolates in diversity hotspots - how did these become isolates; what are alternatives aside from (unsubstantiated) hypothesis 2?

Possibilities for further analysis,

p. 12 "It is plausible to assume, therefore, that the global results arise out of the influence of particular areas of the world, but are not generally applicable."

The current world-wide analysis could be a simple consequence of the distribution of mountains and coastlines around the world. One alternative would be:

Analyze a set of particular areas that exclude parts of the world in which 'peripheral' locations for isolates haven't been relevant. For instance, exclude large 'spread zones', plains without coast access, etc.; take a set amount of km around a mountain range/coast line, analyse all isolates somehow close to it in relation to that mountain range/coast line.

Or work from the other way around, take each isolate and an area around it. Model the isolation in terms of a larger amount of characteristics and their interactions: elevation, inclusion yes/no of access to coasts, how many other languages are there (see below on Voronoi areas). Contrast with non-isolates and not-categorized non-affiliated languages on Glottolog.

Smaller comments.

p. 3 "Normal Diachrony Assumption" this term needs an explanation, or a rephrasing with what it entails very clearly.

p. 5 "requires qualitative information on the historical dynamics of language reduction" - it also requires quantitative information, right. That's just way harder to come by or doesn't exist.

p. 5 "the low power of the test at $p < .05$ by a Spearman's rank correlation" > this sentence doesn't parse well.

p. 7-8. Given the disparity we have in information on European languages vs. all other languages, is there any information on the 'retreat' of Celtic that might be of relevance for the current paper? I am thinking especially of (other peripheral) areas where they could have stayed longer than others; most importantly the Alps.

p. 9; Fig 6 - "whereas for distance to sea, there is also X", I wouldn't put it like this. I would say that the distribution of isolates is less smooth because there are fewer isolates than non-isolates, but that the distributions overlap.

p. 12 "To begin with, there is no sharp qualitative difference between a dialectally diverse isolate and a small language family" I agree with this 100% and would argue for an analysis where each language is modeled primarily in terms of how many family members it has, moving away from the binary "isolate" vs. "non-isolate". (An aggregate measure could be to take the first PCA taking 1) total number of languages in the family; 2) number of subfamilies/genera in the family; 3) number of languages in smallest or intermediate cladistic unit, like Romance within IE). This can be done with Glottolog.

p. 12 "polygon-based datasets of language ranges" - How would they help with the analyses? They would make them more accurate, OK. Any other reason? One solution to this issue would be using Voronoi areas, see:

Harald Hammarström and Tom Güldemann 2014 "Quantifying geographical determinants of large-scale distributions of linguistic features"
& McNew, Derungs, Moran 2018, "Towards faithfully visualizing global linguistic diversity"

p. 12, "The negative result, ..." - Hence, it's quite likely that current measures (distance to mountain range/ distance to sea) don't capture the kind of dynamics that generate isolates. One solution would be to have a quantitative literature-based study on isolates, i.e. gather qualitative information on all isolates in a systematic way in order to compare them quantitatively using more relevant measures, including rugosity but also smth like proximity to major language spreads, and a host of sociolinguistic factors (see for a similar approach Kaius Sinnemäki's GramAdapt project, <https://www2.helsinki.fi/en/researchgroups/linguistic-adaptation/about>).

p. 13 "which many studies of this diversity have yielded so far." sentence stops unexpectedly

p. 13 "For analyses precise definitions" this is a garden path sentence; also the period of this sentence is missing.

p. 15 "The fixed effects have been subject to log10-transformation ..." Why? To reduce the impact of outliers? It would be helpful to get descriptive statistics on the distribution of distances.

p. 15 "New Zealand ... uncoded." Why and how would the results alter if this decision was reverted?

p. 15 Can you give a full spec of the priors, as default priors may change and/or packages may no longer allow default priors in subsequent releases.

p. 15 "To assess reliability of the models given the resulting danger of zero-inflation and overdispersion post hoc, instead, predictive accuracy of the models was assessed following [74]." please provide a section reference, as this resource is not searchable and your reference isn't clear.

Review form: Reviewer 2

Is the manuscript scientifically sound in its present form?

Yes

Are the interpretations and conclusions justified by the results?

Yes

Is the language acceptable?

Yes

Do you have any ethical concerns with this paper?

No

Have you any concerns about statistical analyses in this paper?

No

Recommendation?

Accept with minor revision (please list in comments)

Comments to the Author(s)

I thought this manuscript was a nice example of a robust methodological approach to the complex problem of language isolates. The research setup has a solid approach in evaluating two models, one broadly based on the language diversity and one based on proximity to geographic barriers. I am particularly pleased to see the use of Bayesian Mixed Effects modelling employed in this study, and the reason for this is evident in the manuscript. While the non-parametric tests did tend to show as association between isolates and distance to mountains, the mixed effects model reveals this to be a wrong assumption, presumably because the distance to alpine mountains differs dramatically across the world, as noted by the author. This is precisely the reason why mixed effects models are valuable and it is encouraging to see them used in the domain of language isolates. The author should also be commended for the accessibility of the R code, including code used for the images.

My only concerns about the manuscript is that the author appears to place significant emphasis on explaining and testing the geographic barriers or "isolatogenetic" model with far less discussion on the language diversity model and its associated methods. Even though at the end of the manuscript the author seems to begrudgingly admit that the "strong association between isolates and overall language richness would suggest that whatever drives general language diversification also drives the distribution of isolates." Given this conclusion, the author may want to further develop this model in section 3 and provide a little more context as to the potential mechanisms for how areas of language diversity might encourage isolate development. I want to once again applaud the author for presenting a sort of "negative result" and using it as an opportunity to highlight that two simple models are effective ways to structure a research agenda, but probably too simplistic for the actual processes underlying language isolation. This is especially when moving between global scales of analysis and continental or higher resolution scales.

And perhaps just for clarification, how is the barrier model different than an isolation-by-distance or an isolation-by-environment model, which suggests increase local variation due to limited dispersal or ecological constraints? I bring this up as many readers of RSOS will probably be familiar with “isolation-by” models and it might help to identify how the “isolatogenetic” model relates to these. And perhaps one bit more clarification might be needed here, and this is how to disentangle areas near major barriers with marginal ecological zones. Some of the examples are given as evidence as being near to major barriers, but some of these are presumably located within marginal ecological areas, which are not fully captured in the distance to sea or distance to mountains measurements. As the marginality hypothesis has been discussed previously in the literature, it might be worthwhile to just briefly highlight how you either would take into account ecological productivity, or why ecology can be dismissed as a key factor.

I think a slight restructuring of the manuscript may also want to at least be considered. The most detailed description of two models currently falls in the introduction section (Pg. 3, Lines 1-18). This paragraph might be better served as the first paragraph in section 3, with the last paragraph of the introduction staying as such, although the first sentence of this last paragraph will need to be reworded. This proposed re-structure is not a deal-breaker for the manuscript, but something the author may want to consider. A second consideration would be do more effectively connect the case studies to the models being utilised. For example, it is unclear to me why the Celtic example is even provided, it seems to have little connection with either of the models. From what I gather, these are supposed to be examples of range reduction processes, that are central components of the barriers model? If so, it might be worth foregrounding the concepts of survivor languages as well as spread zones and residual zones prior to the case studies as opposed to after them. The author may even consider a separate section to explain to the models in greater detail (replacing section 3), and then an additional section outlining the historical and qualitative evidence provided in the case studies, with the results section being the quantitative evidence.

Overall, I enjoyed reading this manuscript and was pleased with the methodological approach taken. While the final results showing a broad association at the global level are somewhat unsatisfying, the manuscript demonstrate a well-reasoned approach to testing the models, particularly the barriers model. This manuscript also demonstrates how difficult it is can be to assess diversification processes across different scales and varying degrees of data resolution. I also appreciate the willingness of the author to publish “negative” results and acknowledge that the isolatogenetic model, which they author seems to favour, does not have any “systematic statistical evidence”. Despite this result, this is still a valuable contribution to the literature on language evolution.

Minor Changes

Pg 1. Line 47 Hua et al. 2019 needs to be incorporated into the numbered reference list

There are numerous locations where minor typos exist throughout the paper. Most of these seemed to be in the discussion section and so I would advise a careful review of these paragraphs. Some of the ones I identified here include:

Line 19: Remove “a” between principle and different

Line 36: Remove “underscore this” (perhaps this a note to the author)

Line 38: Insert “the” between expand and scope

Line 39: Remove “likely” as this is redundant with possibility

Line 55: Remove “purely language internal” as it doesn’t make sense in this sentence

Pg. 13, Line 48: Appears to be a random "a" superscripted (perhaps a rendering error?)

Decision letter (RSOS-202232.R0)

Dear Dr Urban

On behalf of the Editors, we are pleased to inform you that your Manuscript RSOS-202232 "The geography and evolution of language isolates" has been accepted for publication in Royal Society Open Science subject to minor revision in accordance with the referees' reports. Please find the referees' comments along with any feedback from the Editors below my signature.

Please submit your revised manuscript and required files (see below) no later than 7 days from today's (ie 11-Mar-2021) date. Note: the ScholarOne system will 'lock' if submission of the revision is attempted 7 or more days after the deadline. If you do not think you will be able to meet this deadline please contact the editorial office immediately.

on behalf of Professor Pete Smith (Subject Editor)
openscience@royalsociety.org

Associate Editor Comments to Author:

Thank you for this submission, which has been received positively by two reviewers. Each recommend a number of modifications, which we'd like you to address - please ensure a full point-by-point response and tracked-changes version of the paper included when you resubmit.

Reviewer comments to Author:

Reviewer: 1

Comments to the Author(s)

The current paper is mostly clearly written (there are a few garden path sentences and typos, and see below on the introduction) and makes a clear and interesting contribution on the geographical distribution of linguistic isolates (not sure it does about 'evolution', though).

This paper has a lot of potential, but suffers slightly from structural issues detailed below. In addition, the analyses it presents are basic, and given the alternatives listed by the author, and the alternatives one thinks of while reading, it feels somehow disappointing that the author hasn't taken the extra steps to do follow through with one or more of these options. However, the work reported on here has to be done before doing more elaborated analyses. I really appreciate the case studies reported on in section 3 and how they lead to hypothesis formation.

Structural issues. I had to read the introduction several times to identify the two opposing views, summarized by:

1: [Isolates] "are generated through the same general historical processes that drives language diversification. As such, it should be the case that linguistic isolates occur in hotspots of linguistic diversity."

2: [Isolates are] "the last surviving member of a former language family [16, 18]. This stance is supported empirically by languages which currently qualify as language isolates, but which are known to have had related members", i.e. we will find isolates in places where they are pushed to following subsequent language spread, "towards a major geographical barrier, prominently the coastline or a major mountain area with conditions that make it not permanently habitable" Even this summary is somehow not as clear as it could be. The main hypotheses on p. 9, "f the account based on the qualitative survey ..." are much clearer.

Conclusion/Spatial Point Pattern Test.

The author concludes (p. 11) that the first hypothesis listed above is right. However, a lot of time is then spend on explaining how hypothesis 2 could still be true. The two hypotheses are not given the same magnitude throughout the paper. Hence, the Spatial Point Pattern Test is not explained at all, nor are alternatives to investigating the relationship between "isolate location and general levels of linguistic diversity on the language level ("language richness". Is the Spatial Point Pattern Test the best possible test? Nor is there much depth given to the processes that would lead to more isolates in diversity hotspots - how did these become isolates; what are alternatives aside from (unsubstantiated) hypothesis 2?

Possibilities for further analysis,

p. 12 "It is plausible to assume, therefore, that the global results arise out of the influence of particular areas of the world, but are not generally applicable."

The current world-wide analysis could be a simple consequence of the distribution of mountains and coastlines around the world. One alternative would be:

Analyze a set of particular areas that exclude parts of the world in which 'peripheral' locations for isolates haven't been relevant. For instance, exclude large 'spread zones', plains without coast access, etc.; take a set amount of km around a mountain range/coast line, analyse all isolates somehow close to it in relation to that mountain range/coast line.

Or work from the other way around, take each isolate and an area around it. Model the isolation in terms of a larger amount of characteristics and their interactions: elevation, inclusion yes/no of access to coasts, how many other languages are there (see below on Voronoi areas). Contrast with non-isolates and not-categorized non-affiliated languages on Glottolog.

Smaller comments.

p. 3 "Normal Diachrony Assumption" this term needs an explanation, or a rephrasing with what it entails very clearly.

p. 5 "requires qualitative information on the historical dynamics of language reduction" - it also requires quantitative information, right. That's just way harder to come by or doesn't exist.

p. 5 "the low power of the test at $p < .05$ by a Spearman's rank correlation" > this sentence doesn't parse well.

p. 7-8. Given the disparity we have in information on European languages vs. all other languages, is there any information on the 'retreat' of Celtic that might be of relevance for the current paper? I am thinking especially of (other peripheral) areas where they could have stayed longer than others; most importantly the Alps.

p. 9; Fig 6 - "whereas for distance to sea, there is also X", I wouldn't put it like this. I would say that the distribution of isolates is less smooth because there are fewer isolates than non-isolates, but that the distributions overlap.

p. 12 "To begin with, there is no sharp qualitative difference between a dialectally diverse isolate and a small language family" I agree with this 100% and would argue for an analysis where each language is modeled primarily in terms of how many family members it has, moving away from the binary "isolate" vs. "non-isolate". (An aggregate measure could be to take the first PCA taking 1) total number of languages in the family; 2) number of subfamilies/genera in the family; 3) number of languages in smallest or intermediate cladistic unit, like Romance within IE). This can be done with Glottolog.

p. 12 "polygon-based datasets of language ranges" - How would they help with the analyses? They would make them more accurate, OK. Any other reason? One solution to this issue would be using Voronoi areas, see:

Harald Hammarström and Tom Güldemann 2014 "Quantifying geographical determinants of large-scale distributions of linguistic features"
& McNew, Derungs, Moran 2018, "Towards faithfully visualizing global linguistic diversity"

p. 12, "The negative result, ..." - Hence, it's quite likely that current measures (distance to mountain range/distance to sea) don't capture the kind of dynamics that generate isolates. One solution would be to have a quantitative literature-based study on isolates, i.e. gather qualitative information on all isolates in a systematic way in order to compare them quantitatively using more relevant measures, including rugosity but also smth like proximity to major language spreads, and a host of sociolinguistic factors (see for a similar approach Kaius Sinnemäki's GramAdapt project, <https://www2.helsinki.fi/en/researchgroups/linguistic-adaptation/about>).

p. 13 "which many studies of this diversity have yielded so far." sentence stops unexpectedly

p. 13 "For analyses precise definitions" this is a garden path sentence; also the period of this sentence is missing.

p. 15 "The fixed effects have been subject to log10-transformation ..." Why? To reduce the impact of outliers? It would be helpful to get descriptive statistics on the distribution of distances.

p. 15 "New Zealand ... uncoded." Why and how would the results alter if this decision was reverted?

p. 15 Can you give a full spec of the priors, as default priors may change and/or packages may no longer allow default priors in subsequent releases.

p. 15 "To assess reliability of the models given the resulting danger of zero-inflation and overdispersion post hoc, instead, predictive accuracy of the models was assessed following [74]." please provide a section reference, as this resource is not searchable and your reference isn't clear.

Reviewer: 2

Comments to the Author(s)

I thought this manuscript was a nice example of a robust methodological approach to the complex problem of language isolates. The research setup has a solid approach in evaluating two models, one broadly based on the language diversity and one based on proximity to geographic barriers. I am particularly pleased to see the use of Bayesian Mixed Effects modelling employed in this study, and the reason for this is evident in the manuscript. While the non-parametric tests did tend to show an association between isolates and distance to mountains, the mixed effects model reveals this to be a wrong assumption, presumably because the distance to alpine mountains differs dramatically across the world, as noted by the author. This is precisely the reason why mixed effects models are valuable and it is encouraging to see them used in the domain of language isolates. The author should also be commended for the accessibility of the R code, including code used for the images.

My only concerns about the manuscript is that the author appears to place significant emphasis on explaining and testing the geographic barriers or "isolatogenetic" model with far less discussion on the language diversity model and its associated methods. Even though at the end of the manuscript the author seems to begrudgingly admit that the "strong association between isolates and overall language richness would suggest that whatever drives general language diversification also drives the distribution of isolates." Given this conclusion, the author may want to further develop this model in section 3 and provide a little more context as to the potential mechanisms for how areas of language diversity might encourage isolate development. I want to once again applaud the author for presenting a sort of "negative result" and using it as an opportunity to highlight that two simple models are effective ways to structure a research agenda, but probably too simplistic for the actual processes underlying language isolation. This is especially when moving between global scales of analysis and continental or higher resolution scales.

And perhaps just for clarification, how is the barrier model different than an isolation-by-distance or an isolation-by-environment model, which suggests increase local variation due to limited dispersal or ecological constraints? I bring this up as many readers of RSOS will probably be familiar with "isolation-by-" models and it might help to identify how the "isolatogenetic" model relates to these. And perhaps one bit more clarification might be needed here, and this is how to disentangle areas near major barriers with marginal ecological zones. Some of the examples are given as evidence as being near to major barriers, but some of these are presumably located within marginal ecological areas, which are not fully captured in the distance to sea or distance to mountains measurements. As the marginality hypothesis has been discussed previously in the literature, it might be worthwhile to just briefly highlight how you either would take into account ecological productivity, or why ecology can be dismissed as a key factor.

I think a slight restructuring of the manuscript may also want to at least be considered. The most detailed description of two models currently falls in the introduction section (Pg. 3, Lines 1-18).

This paragraph might be better served as the first paragraph in section 3, with the last paragraph of the introduction staying as such, although the first sentence of this last paragraph will need to be reworded. This proposed re-structure is not a deal-breaker for the manuscript, but something the author may want to consider. A second consideration would be to do more effectively connect the case studies to the models being utilised. For example, it is unclear to me why the Celtic example is even provided, it seems to have little connection with either of the models. From what I gather, these are supposed to be examples of range reduction processes, that are central components of the barriers model? If so, it might be worth foregrounding the concepts of survivor languages as well as spread zones and residual zones prior to the case studies as opposed to after them. The author may even consider a separate section to explain the models in greater detail (replacing section 3), and then an additional section outlining the historical and qualitative evidence provided in the case studies, with the results section being the quantitative evidence.

Overall, I enjoyed reading this manuscript and was pleased with the methodological approach taken. While the final results showing a broad association at the global level are somewhat unsatisfying, the manuscript demonstrates a well-reasoned approach to testing the models, particularly the barriers model. This manuscript also demonstrates how difficult it can be to assess diversification processes across different scales and varying degrees of data resolution. I also appreciate the willingness of the author to publish "negative" results and acknowledge that the isolatogenetic model, which they author seems to favour, does not have any "systematic statistical evidence". Despite this result, this is still a valuable contribution to the literature on language evolution.

Minor Changes

Pg 1. Line 47 Hua et al. 2019 needs to be incorporated into the numbered reference list

There are numerous locations where minor typos exist throughout the paper. Most of these seemed to be in the discussion section and so I would advise a careful review of these paragraphs. Some of the ones I identified here include:

Line 19: Remove "a" between principle and different

Line 36: Remove "underscore this" (perhaps this a note to the author)

Line 38: Insert "the" between expand and scope

Line 39: Remove "likely" as this is redundant with possibility

Line 55: Remove "purely language internal" as it doesn't make sense in this sentence

Pg. 13, Line 48: Appears to be a random "a" superscripted (perhaps a rendering error?)

===PREPARING YOUR MANUSCRIPT===

===PREPARING YOUR REVISION IN SCHOLARONE===

-- Ensure that your data access statement meets the requirements at <https://royalsociety.org/journals/authors/author-guidelines/#data>. You should ensure that you cite the dataset in your reference list. If you have deposited data etc in the Dryad repository, please only include the 'For publication' link at this stage. You should remove the 'For review' link.

Author's Response to Decision Letter for (RSOS-202232.R0)

See Appendix A.

Decision letter (RSOS-202232.R1)

Dear Dr Urban,

I am pleased to inform you that your manuscript entitled "The geography and development of language isolates" is now accepted for publication in Royal Society Open Science.

You can expect to receive a proof of your article in the near future. Please contact the editorial office (openscience@royalsociety.org) and the production office (openscience_proofs@royalsociety.org) to let us know if you are likely to be away from e-mail contact -- if you are going to be away, please nominate a co-author (if available) to manage the proofing process, and ensure they are copied into your email to the journal. Due to rapid

publication and an extremely tight schedule, if comments are not received, your paper may experience a delay in publication.

on behalf of Prof Pete Smith (Subject Editor)
openscience@royalsociety.org

Appendix A

General Response

Both reviews I received for this manuscript are detailed, provide many helpful comments and suggestions, and show that the reviewers read the paper very carefully. I would like to thank both reviewers sincerely for providing such high quality reviews.

The reviewers converge on some points that require attention.

*Reviewer 1 notes some issues with **clarity in the introductory section**, and relatedly, Reviewer 2 suggests some restructuring of content with respect to this and the following section of texts. That both reviewers make pertinent remarks here shows that there is indeed the need to improve on the text. After some experimenting, I have not taken up Reviewer 2's suggestion to move parts of the introduction to the following section, but rather have left an improved version of the text in the introduction to create a more clearly articulated statement of the two opposing views in one and the same section (as carved out by reviewer 1).*

*Second, they both note an **imbalance with regard to the discussion of the two explanatory mechanisms the article considers**, -isolates as a general phenomenon of linguistic diversity and as survivors of former language families near geographical barriers- both in terms of the length of the discussion devoted to each and also the statistical attention they receive. It is somewhat inevitable that the latter receives somewhat longer discussion because it has to be developed through case studies. But it is true that more space and attention could also be devoted to developing the former. I have slightly rewritten the introduction integrated the evidence from the Upper Amazon as an example of a possible process that, in the long run, is capable generating both general language richness and isolate richness.*

*Third, the reviewers both make suggestions as to **conceptual and statistical refinements**. Such added analyses would require a careful design and further assessment. I point out more clearly the limitations of this article and integrate many of the suggestions for improvement into the final discussion, attributing them to the reviewers.*

More point-by-point responses to the reviewers' comments follow interspersed with their text.

Reviewer: 1

Comments to the Author(s)

The current paper is mostly clearly written (there are a few garden path sentences and typos, and see below on the introduction) and makes a clear and interesting contribution on the geographical distribution of linguistic isolates (not sure it does about 'evolution', though).

I actually agree with the reviewer that the term "evolution" is a bit overused in recent quantitative approaches to linguistic diversity. I have therefore replaced "evolution" with the term "development" or similar more neutral terms.

This paper has a lot of potential, but suffers slightly from structural issues detailed below. In addition, the analyses it presents are basic, and given the alternatives listed by the author, and the alternatives one thinks of while reading, it feels somehow disappointing that the author hasn't taken the extra steps to do follow through with one or more of these options. However, the work reported on here has to be done before doing more elaborated analyses. I really appreciate the case studies reported on in section 3 and how they lead to hypothesis formation.

Thank you, I am glad that you appreciate the case studies. See the general response for possible refinements.

Structural issues. I had to read the introduction several times to identify the two opposing views, summarized by:

1: [Isolates] "are generated through the same general historical processes that drives language diversification. As such, it should be the case that linguistic isolates occur in hotspots of linguistic diversity."

2: [Isolates are] "the last surviving member of a former language family [16, 18]. This stance is supported empirically by languages which currently qualify as language isolates, but which are known to have had related members", i.e. we will find isolates in places where they are pushed to following subsequent language spread, "towards a major geographical barrier, prominently the coastline or a major mountain area with conditions that make it not permanently habitable"

Even this summary is somehow not as clear as it could be. The main hypotheses on p. 9, "f the account based on the qualitative survey ..." are much clearer.

I have thoroughly rewritten the introduction, hoping that the opposing views are more clearly articulated. See also the general response at the beginning of this document.

Conclusion/Spatial Point Pattern Test.

The author concludes (p. 11) that the first hypothesis listed above is right. However, a lot of time is then spend on explaining how hypothesis 2 could still be true. The two hypotheses are not given the same magnitude throughout the paper. Hence, the Spatial Point Pattern Test is not explained at all, nor are alternatives to investigating the relationship between "isolate location and general levels of linguistic diversity on the language level ("language richness")." Is the Spatial Point Pattern Test the best possible test? Nor is there much depth given to the processes that would lead to more isolates in diversity hotspots - how did these become isolates; what are alternatives aside from (unsubstantiated) hypothesis 2?

I have added some more explanation of the Spatial Point Pattern Test (which is explained in the "Data and Method" section) also to the main text. See the general response for the imbalance between the two hypotheses as discussed in the paper and the absence of discussion of "processes that would lead to more isolates in diversity hotspots". I have mitigated this imbalance, and discuss one such process.

Possibilities for further analysis,

p. 12 "It is plausible to assume, therefore, that the global results arise out of the influence of particular areas of the world, but are not generally applicable."

The current world-wide analysis could be a simple consequence of the distribution of mountains and coastlines around the world. One alternative would be:

Analyze a set of particular areas that exclude parts of the world in which 'peripheral' locations for isolates haven't been relevant. For instance, exclude large 'spread zones', plains without coast access, etc.; take a set amount of km around a mountain range/coast line, analyse all isolates somehow close to it in relation to that mountain range/coast line.

Or work from the other way around, take each isolate and an area around it. Model the isolation in terms of a larger amount of characteristics and their interactions: elevation, inclusion yes/no of access to coasts, how many other languages are there (see below on Voronoi areas). Contrast with non-isolates and not-categorized non-affiliated languages on Glottolog.

I mention these analytic possibilities in the concluding section, thanks. I also include a reference to recent work where a similar approach has actually been taken already.

Smaller comments.

p. 3 "Normal Diachrony Assumption" this term needs an explanation, or a rephrasing with what it entails very clearly.

I have added additional explanations.

p. 5 "requires qualitative information on the historical dynamics of language reduction" - it also requires quantitative information, right. That's just way harder to come by or doesn't exist.

Ultimately, yes. For the purpose of hypothesis formation which is at stake in the context of this statement (and which the reviewers highlights as praiseworthy further above), qualitative case studies are sufficient. I therefore prefer to leave the text unchanged.

p. 5 "the low power of the test at p this sentence doesn't parse well.

I have rephrased.

p. 7-8. Given the disparity we have in information on European languages vs. all other languages, is there any information on the 'retreat' of Celtic that might be of relevance for the current paper? I am thinking especially of (other peripheral) areas where they could have stayed longer than others; most importantly the Alps.

This is a good point. In fact, I had looked into the dynamics of the Celtic "retreat" already when writing the original submission, but found surprisingly little information. This may have something to do with the fact that, even though we have much more information on the linguistic history of Europe, the Continental Celtic languages are only documented in short inscriptions, graffiti, etc. So even the knowledge of the languages themselves is very patchy. I have added what little information I could find in the literature to the text.

p. 9; Fig 6 - "whereas for distance to sea, there is also X", I wouldn't put it like this. I would say that the distribution of isolates is less smooth because there are fewer isolates than non-isolates, but that the distributions overlap.

I have rephrased.

p. 12 "To begin with, there is no sharp qualitative difference between a dialectally diverse isolate and a small language family" I agree with this 100% and would argue for an analysis where each language is modeled primarily in terms of how many family members it has, moving away from the binary "isolate" vs. "non-isolate". (An aggregate measure could be to take the first PCA taking 1) total number of languages in the family; 2) number of subfamilies/genera in the family; 3) number of languages in smallest or intermediate cladistic unit, like Romance within IE). This can be done with Glottolog.

This is also a good suggestion, which I take up in the concluding section (attributing it to the reviewer).

p. 12 "polygon-based datasets of language ranges" - How would they help with the analyses? They would make them more accurate, OK. Any other reason? One solution to this issue would be using Voronoi areas, see:

Harald Hammarström and Tom Güldemann 2014 "Quantifying geographical determinants of large-scale distributions of linguistic features"
& McNew, Derungs, Moran 2018, "Towards faithfully visualizing global linguistic diversity"

I have added clarification for how polygon-based datasets would allow to improve analyses, in particular by eliminating uncertainties associated with point-based representations of languages.

p. 12, "The negative result, ..." - Hence, it's quite likely that current measures (distance to mountain range/distance to sea) don't capture the kind of dynamics that generate isolates. One solution would be to have a quantitative literature-based study on isolates, i.e. gather qualitative information on all isolates in a systematic way in order to compare them quantitatively using more relevant measures, including rugosity but also smth like proximity to major language spreads, and a host of sociolinguistic factors (see for a similar approach Kaius Sinnemäki's GramAdapt project, <https://www2.helsinki.fi/en/researchgroups/linguistic-adaptation/about>).

I agree, and have added discussion along these lines to the final section of the article.

p. 13 "which many studies of this diversity have yielded so far." sentence stops unexpectedly

Thank you. I have rephrased.

p. 13 "For analyses precise definitions" this is a garden path sentence; also the period of this sentence is missing.

Thank you. I have rephrased.

p. 15 "The fixed effects have been subject to log10-transformation ..." Why? To reduce the impact of outliers? It would be helpful to get descriptive statistics on the distribution of distances.

This was done "to reduce skew" (as the original submission actually already states). Fig. 6 provides an accessible visual representation of the distributions.

p. 15 "New Zealand ... uncoded." Why and how would the results alter if this decision was reverted?

This is a fair question. I have run a model that retains the "Papunesia" macro-area and otherwise had the same specifications as the model of the main analysis. The answer to the question is that the results are altered only negligibly, and that no different conclusions would be drawn from a model that retains the "Papunesia" area. This is reported in the "data and

methods section”, with full results of this ancillary model in Supplementary Table S2.

p. 15 Can you give a full spec of the priors, as default priors may change and/or packages may no longer allow default priors in subsequent releases.

Yes, this information is now included in the “Data and Methods” section

p. 15 "To assess reliability of the models given the resulting danger of zero-inflation and overdispersion post hoc, instead, predictive accuracy of the models was assessed following [74]." please provide a section reference, as this resource is not searchable and your reference isn't clear.

Done.

Reviewer: 2

Comments to the Author(s)

I thought this manuscript was a nice example of a robust methodological approach to the complex problem of language isolates. The research setup has a solid approach in evaluating two models, one broadly based on the language diversity and one based on proximity to geographic barriers. I am particularly pleased to see the use of Bayesian Mixed Effects modelling employed in this study, and the reason for this is evident in the manuscript. While the non-parametric tests did tend to show an association between isolates and distance to mountains, the mixed effects model reveals this to be a wrong assumption, presumably because the distance to alpine mountains differs dramatically across the world, as noted by the author. This is precisely the reason why mixed effects models are valuable and it is encouraging to see them used in the domain of language isolates. The author should also be commended for the accessibility of the R code, including code used for the images.

My only concerns about the manuscript is that the author appears to place significant emphasis on explaining and testing the geographic barriers or “isolatogenetic” model with far less discussion on the language diversity model and its associated methods. Even though at the end of the manuscript the author seems to begrudgingly admit that the “strong association between isolates and overall language richness would suggest that whatever drives general language diversification also drives the distribution of isolates.” Given this conclusion, the author may want to further develop this model in section 3 and provide a little more context as to the potential mechanisms for how areas of language diversity might encourage isolate development. I want to once again applaud the author for presenting a sort of “negative result” and using it as an opportunity to highlight that two simple models are effective ways to structure a research agenda, but probably too simplistic for the actual processes underlying language isolation. This is especially when moving between global scales of analysis and continental or higher resolution scales.

I take the point regarding the imbalance between the elaborateness of the discussion of both models, and have taken measures to mitigate this. See also the second item in the general response at the beginning of this document.

And perhaps just for clarification, how is the barrier model different than an isolation-by-distance or an isolation-by-environment model, which suggests increase local variation due to limited dispersal or ecological constraints? I bring this up as many readers of RSOS will

probably be familiar with “isolation-by” models and it might help to identify how the “isolatogenetic” model relates to these.

Thank you for this suggestion. I have added a paragraph that contextualizes the model with “isolation-by” models. It seems that isolation-by-environment is the closest analogue here. I also discuss refugial isolation.

And perhaps one bit more clarification might be needed here, and this is how to disentangle areas near major barriers with marginal ecological zones. Some of the examples are given as evidence as being near to major barriers, but some of these are presumably located within marginal ecological areas, which are not fully captured in the distance to sea or distance to mountains measurements. As the marginality hypothesis has been discussed previously in the literature, it might be worthwhile to just briefly highlight how you either would take into account ecological productivity, or why ecology can be dismissed as a key factor.

I think this suggestion hits the nail on the head, and I have incorporated in particular the reviewers remark that ecological marginality is “not fully captured in the distance to sea or distance to mountains measurements” into section 6.

I think a slight restructuring of the manuscript may also want to at least be considered. The most detailed description of two models currently falls in the introduction section (Pg. 3, Lines 1-18). This paragraph might be better served as the first paragraph in section 3, with the last paragraph of the introduction staying as such, although the first sentence of this last paragraph will need to be reworded. This proposed re-structure is not a deal-breaker for the manuscript, but something the author may want to consider.

See the first point in the general response. I have chosen to rather follow up on Reviewer 1 and have rewritten the introduction accordingly. I hope, however, that this served to increase clarity of the text, which reviewer 2 must also have had in mind when suggesting this restructuring, and that therefore the intended goal is achieved though on a different route.

A second consideration would be do more effectively connect the case studies to the models being utilised. For example, it is unclear to me why the Celtic example is even provided, it seems to have little connection with either of the models. From what I gather, these are supposed to be examples of range reduction processes, that are central components of the barriers model? If so, it might be worth foregrounding the concepts of survivor languages as well as spread zones and residual zones prior to the case studies as opposed to after them. The author may even consider a separate section to explain to the models in greater detail (replacing section 3), and then an additional section outlining the historical and qualitative evidence provided in the case studies, with the results section being the quantitative evidence.

Thank you for the comment on Celtic. Indeed, I haven't made explicit enough why this is relevant to the model. I have added explanations to the text that should make things clearer. I have seriously considered rearranging the content of the text in the way the reviewer suggests. In the end found it preferable to leave it more or less as it is, because this reflects more closely how the formulation of the hypothesis and the model of the genesis of isolates actually happened, and I find it beneficial if the reader can retrace that (note that reviewer 1 agrees when saying that they “really appreciate the case studies reported on in section 3 and how they lead to hypothesis formation”).

Overall, I enjoyed reading this manuscript and was pleased with the methodological approach

taken. While the final results showing a broad association at the global level are somewhat unsatisfying, the manuscript demonstrate a well-reasoned approach to testing the models, particularly the barriers model. This manuscript also demonstrates how difficult it is can be to assess diversification processes across different scales and varying degrees of data resolution. I also appreciate the willingness of the author to publish “negative” results and acknowledge that the isolatogenetic model, which they author seems to favour, does not have any “systematic statistical evidence”. Despite this result, this is still a valuable contribution to the literature on language evolution.

Thank you very much.

Minor Changes

Pg 1. Line 47 Hua et al. 2019 needs to be incorporated into the numbered reference list

Thank you for catching this. Done.

There are numerous locations where minor typos exist throughout the paper. Most of these seemed to be in the discussion section and so I would advise a careful review of these paragraphs. Some of the ones I identified here include:

Line 19: Remove “a” between principle and different

Line 36: Remove “underscore this” (perhaps this a note to the author)

Line 38: Insert “the” between expand and scope

Line 39: Remove “likely” as this is redundant with possibility

Line 55: Remove “purely language internal” as it doesn’t make sense in this sentence

Pg. 13, Line 48: Appears to be a random “a” superscripted (perhaps a rendering error?)

Thank you very much. All fixed.

===PREPARING YOUR MANUSCRIPT===

While not essential, it will speed up the preparation of your manuscript proof if you format your references/bibliography in Vancouver style (please see <https://royalsociety.org/journals/authors/author-guidelines/#formatting>). **You should include DOIs for as many of the references as possible.**

===PREPARING YOUR REVISION IN SCHOLARONE===
